# Physical, Mechanical and Biological Properties of Phenolic Acid-Grafted Soluble Soybean Polysaccharide Films

**DOI:** 10.3390/foods11223747

**Published:** 2022-11-21

**Authors:** Mengyang Zhang, Chen Huang, Jing Xie, Zehuai Shao, Xiaohui Li, Xiaojun Bian, Bin Xue, Jianhong Gan, Tao Sun

**Affiliations:** 1Shanghai Engineering Research Center of Aquatic-Product Processing & Preservation, Shanghai 201306, China; 2College of Food Science & Technology, Shanghai Ocean University, Shanghai 201306, China; 3Quality Supervision, Inspection & Testing Center for Cold Storage and Refrigeration Equipment, Ministry of Agriculture, Shanghai 201306, China

**Keywords:** soluble soybean polysaccharide, phenolic acid, grafted, film

## Abstract

Three kinds of phenolic acid-grafted soluble soybean polysaccharide (SSPS) with similar grafting ratios were prepared, and their structure was characterized by FT-IR, UV-vis and ^1^ H NMR. The impact of phenolic acid on the antioxidant activity of SSPS was evaluated. Then, films were prepared by using phenolic acid-grafted SSPS. The physical, mechanical and biological performances of phenolic acid-grafted SSPS films were further investigated. The results indicated that an ester linkage was formed between the SSPS and phenolic acid. The grafting ratio of para-hydroxybenzoic acid, protocatechuic acid and gallic acid-grafted SSPS was 29.45, 31.76 and 30.74 mg/g, respectively. Phenolic acid endowed SSPS with improved antioxidant properties. Gallic acid (GA)-grafted SSPS possessed the best DPPH radical scavenging ability and reducing power, which may be related to the three phenolic hydroxyl groups in GA. Phenolic acid-grafted SSPS films showed increased moisture content and decreased water solubility compared to SSPS film. The phenolic acid-g-SSPS decreased the mechanical properties but enhanced the water vapor barrier property, and antioxidant and antibacterial properties of SSPS film. Meanwhile, the para-hydroxybenzoic acid-grafted SSPS film showed the lowest water vapor permeability (3.70 × 10^−7^ g mm/h cm^2^ Pa), and the GA-grafted SSPS film exhibited the best antioxidant and antibacterial activities.

## 1. Introduction

Petroleum-derived plastic materials are excessively consumed and are difficult to degrade, causing serious environmental pollution [1]. Thus, environmentally friendly biological-based materials, such as lipids, proteins, and polysaccharides, that can replace plastic materials are becoming more and more popular. These bio-based materials are non-toxic, biodegradable and biocompatible and have been developed into edible films, which have broad application prospects in the food industry [2].

Soluble soybean polysaccharides having a structural resemblance to pectin is an acidic polysaccharide. SSPS possesses a galacturonan skeleton of rhamnogalacturonan and homogalacturonan (α-1,4-galacturonan) with a side chain of α-1,3- or α-1,5-arabinan and β-1,4-galactan [3]. SSPS exhibits high solubility, low viscosity, emulsifying properties, thermal stability, is biodegradable and has a good film-forming capability [4,5,6]. Previous studies have revealed that SSPS films exhibited good mechanical properties and appearance [7,8,9], but SSPS films did not show satisfactory antioxidant and antimicrobial activities. Therefore, the application of SSPS in food packaging can be greatly improved by enhancing the antioxidant and antibacterial activities of SSPS.

In recent years, chitosan (CS) has been frequently grafted with various phenolic acids to enhance its water solubility and antioxidant and antibacterial activities [10,11], which endowed CS with favorable biological activity and enhanced CS-based films’ functionality [12]. Protocatechuic acid (PA)-grafted carboxymethyl chitosan showed drastic enhancements in antioxidant ability [13]. PA-grafted CS film showed better performances than CS film, such as improved UV-vis light barrier properties, mechanical properties and antioxidant ability [14]. GA-grafted CS improved the biological abilities of CS film, and the abilities were positively correlated with the grafting ratio [15]. Compared with CS film, phenolic acid-grafted CS film showed decreased WVP, increased mechanical performance and enhanced antioxidant ability, and GA-grafted CS film had the best antioxidant activity because of its highest grafting ratio and three phenolic hydroxyl groups [2]. The performances of phenolic acid-grafted CS films depend on the grafting ratio and the structure of phenolic acid. Nevertheless, the impact of phenolic acid on SSPS and SSPS films has never been studied before.

In this work, three kinds of phenolic acid-grafted SSPS (phenolic acid-g-SSPS) with similar grafting ratios were prepared. Structural characterization of phenolic acid-g-SSPS was carried out. Then, the phenolic acid-g-SSPS was further prepared into films. Finally, the physical and mechanical properties and biological performances of phenolic acid-g-SSPS films were studied.

## 2. Materials and Methods

### 2.1. Materials

Soluble soybean polysaccharide (SSPS, Mw: 160 kDa), para-hydroxybenzoic acid (PPA), gallic acid (GA), protocatechuic acid (PA), 1,1-diphenyl-2-picrylhydrazyl (DPPH), gelatin, Folin-Ciocalteu-reagent, N-hydroxysuccinimide (NHS) and 1-Ethyl-3-(3-dimethylaminopropyl) carbodiimide hydrochloride (EDC) were all bought from Yuanye Biological Co., Ltd., Shanghai, China.

### 2.2. Preparation of Phenolic Acid-g-SSPS

A series of phenolic acid-g-SSPSs were prepared, and the three kinds of phenolic acid-g-SSPSs with similar grafting ratios were selected for further evaluation. Details are as follows. 

Phenolic acid-g-SSPS was synthesized by the EDC/NHS reaction according to Shreiber’s method [16]. SSPS (2.0 g, 0.33 mmol)) was completely dispersed in deionized water (100 mL, 50 ℃). Then, GA (2.20 g, 12.94 mmol) was evenly mixed with aqueous ethanol (20 mL, 70%, *v*/*v*), followed by the addition of EDC (2.20 g, 11.48 mmol). Then, NHS (2.20 g, 19.12 mmol) was added to the reaction system and stirred for 2 h in the ice bath (4 °C). Afterward, the reactive system was mixed with the prepared SSPS solution, and the solution was stirred at room temperature for 48 h after stirring in the ice bath for 4 h. Then, the reaction solution was centrifuged (1 × 10^4^ r/min, 35 min) to remove residual GA. The final product was obtained by dialyzing against distilled water for a three-day period (cut-off membrane: 14,000 Da), lyophilized for 24 h and named GA-g-SSPS. Similarly, PPA-g-SSPS and PA-g-SSPS were synthesized according to the aforementioned method when the PPA (2.20 g, 15.94 mmol) and PA (1.10 g, 7.14 mmol) were substituted for GA. The yields of PPA-g-SSPS, PA-g-SSPS and GA-g-SSPS were 70.59%, 51.70% and 58.53%, respectively.

### 2.3. Structure Analysis of Phenolic Acid-g-SSPS

#### 2.3.1. Fourier Transform-Infrared Spectrometry (FT-IR)

FT-IR spectra of phenolic acid-g-SSPS were recorded on a Nicolet Is 10 FT-IR spectrometer (Thermo Fisher Scientific Co., Ltd., Shanghai, China) from 600–4000 cm^−1^ at a resolution of 4 cm^−1^ [17]. 

#### 2.3.2. UV-Vis Spectroscopy

UV-vis spectra of sample (1 mg/mL) were recorded (200–400 nm) using a U-3900 spectrophotometer (Hitachi (China) Ltd., Tokyo, Japan) [18].

#### 2.3.3. Proton Nuclear Magnetic Resonance Spectroscopy (^1^ H NMR)

The sample (10 mg) was completely dissolved in D_2_O (0.55 mL). Then, ^1^ H NMR spectra were recorded at 25 °C at 400 MHz by using a Bruker Avance III HD spectrometer (Bruker Inc., Ettlingen, Germany) [19].

#### 2.3.4. The Grafting Ratio of Phenolic Acid-g-SSPS

Folin-Ciocalteu reagent (1 mL, 1 mol/L) and phenolic acid-g-SSPS sample (1 mL, 1 mg/mL) were mixed, and the reaction was kept at 30 ℃ for 5 min in the darkness. Then, 5 mL of Na_2_CO_3_ solution (20%, *w*/*v*) and deionized water (3 mL) was added, and the reaction was continued for 2 h in a dark environment. At the final step, the absorbance value at 747 nm was read. The grafting ratio was determined by the standard curve of phenolic acid and indicated as phenolic acid (mg)/dry weight of the sample (g) [20]. 

### 2.4. Antioxidant Activity of Phenolic Acid-g-SSPS

#### 2.4.1. DPPH Radical Scavenging Ability

Referring to Liu’s method [21] with slight modifications, the DPPH radical scavenging ability of phenolic acid-g-SSPS was evaluated. The DPPH ethanol solution (3.0 mL, 0.1 mmol/L) and sample solution (1.0 mL) were mixed. Under dark and room temperature conditions, the above reaction was conducted for half an hour. Its absorbance was recorded at 517 nm to calculate the DPPH scavenging activity:(1)$$\mathrm{DPPH}\;\mathrm{scavenging}\;\mathrm{activity}\;(\%)=\left(1-\frac{A_{sample}-A_{sample\;only}}{A_{control}}\right)\;\times \;100\;$$

#### 2.4.2. Reducing Power

The reducing power was tested according to Oyaizu’s method [22]. Phenolic acid-g-SSPS solution (2.5 mL), K_3_[Fe(CN)_6_] reagent (2.5 mL, 1%, *w*/*v*) and sodium phosphate buffer solution (2.5 mL, pH 6.6) were mixed; then the mixture was incubated in water (50 °C, 20 min) after evenly mixing in a vortex oscillator. Trichloroacetic acid (TCA) (0.5 mL, 10%, *w*/*v*) was mixed with the reaction solution and centrifuged for 10 min. The supernatant of the mixture (2.5 mL), deionized water (2.5 mL) and FeCl_3_ solution (0.5 mL, 0.1%, *w*/*v*) were mixed, and the mixture was allowed to stand for 0.5 h. At last, the absorbance was obtained at 700 nm.

### 2.5. Preparation of Phenolic Acid-g-SSPS Films

Phenolic acid-g-SSPS films were obtained in accordance with Liu’s way with some changes [23]. Phenolic acid-g-SSPS (4.0 g) and gelatin (1.0 g) were solubilized in water (100 mL) with stirring at 60 °C (800 rpm, 40 min), then glycerol (0.96 g) was put into the solution. The film-forming solution was stirred for 15 min at 85 °C and degassed by sonication. Then, 80 mL of the film solution was evenly dispersed onto a 30 × 20 cm polycarbonate sheet. It was left to dry at 25 °C for 2 days, and the dried film was removed and placed in a constant humidity box (25 °C, 50% relative humidity) for two days before assessment. The SSPS film was prepared as a control when SSPS (4 g) was substituted for phenolic acid-g-SSPS.

### 2.6. Physical and Mechanical Properties

#### 2.6.1. Thickness

A spiral micrometer (Nscing Es 217-111, Nanjing, China) was used to test the thickness of films at any of 10 locations (as dispersed as possible) [24].

#### 2.6.2. Moisture Content and Water Solubility 

The sample film was cut into 3 × 3 cm squares and weighed on an analytical balance (*W*_0_), then the film was placed in an oven (105 °C) and dried to a constant weight (*W*_1_). Then, the film was soaked in 50 mL of deionized water and stirred slowly for 6 h. The remaining film was removed and placed into the oven again, and dried to a stable weight (*W*_2_) [25]. The moisture content (%) and water solubility (%) were calculated by the following equations.
(2)$$\mathrm{Moisture}\;\mathrm{content}\;(\%)=\frac{W_{0}-W_{1}}{W_{0}}\;\times \;100\;$$
(3)$$\mathrm{Water}\;\mathrm{solubility}\;(\%)=\frac{W_{1}-W_{2}}{W_{1}}\;\times \;100\;$$

#### 2.6.3. Water Vapor Permeability (WVP)

The WVP of the film was measured by using the method of Gomez-Estaca [26]. The film sample was sliced into 6 × 6 cm squares, and then the color-changing silica gel was placed into a 30 × 60 mm weighing bottle, maintaining 1 cm spacing between the mouth of the bottle and the silica gel. Then the film was tightly covered over, and the mouth of the weighing bottle was tied with a rubber band, which was put into a glass desiccator with deionized water in the base. Finally, weighing was performed every 2 h for 1 day.
(4)$$\mathrm{Water}\;\mathrm{vapor}\;\mathrm{permeability}=\frac{M\times d}{t\times A\times \Delta \;P}\;(g\;\mathrm{mm}/h\;{\mathrm{cm}}^{2}\;\mathrm{Pa})\;$$
where *M* indicates the increasing weight of the weighing bottle (g), *d* indicates the thickness (mm), *t* indicates the time period of weight measurement (h), *A* indicates the circular area covering the mouth of the weighing bottle (cm^2^) and ΔP indicates the difference in vapor pressure between the outside and inside of the film (3567 Pa at 27 °C).

#### 2.6.4. Mechanical Properties

The mechanical performance of the film was evaluated on a DCP-KZ300 experimental machine (Ming Chi Instrument Co. Ltd., Chengdu, China). The rectangular film (150 × 15 mm) was placed on the measuring equipment with a spacing of 60 mm and a stretching speed of 50 mm/min [27]. The tensile strength (TS) and elongation at break (EB) of the film were calculated as follows:(5)$$\mathrm{Tensile}\;\mathrm{strength}\;(\mathrm{MPa})=\frac{F}{d\times h}$$
(6)$$\mathrm{Elongation}\;\mathrm{at}\;\mathrm{break}\;(\%)=\frac{\Delta \;L}{L_{0}}\;\;\times \;100\;$$
where *F* was the stress at film break (N), *d* was the film thickness (mm), *h* was the width of the film (mm), ΔL was the increase in length at the breakpoint (mm) and *L_0_* was the initial length of the film (mm).

### 2.7. The Released Phenol Content and DPPH Radical Scavenging Activity

The released phenol content and DPPH radical scavenging ability were determined by Liu’s method [28]. Briefly, 40 mg of the film was soaked in 4 mL of deionized water for 1 day. Afterward, the released phenol content in supernatant (1.0 mL) was measured (same as Section 2.3.4). In addition, the supernatant (0.1 mL) was mixed with DPPH ethanol solution (5.0 mL, 0.1 mmol/L) to measure the DPPH radical scavenging ability of the film, as mentioned in Section 2.4.1. 

### 2.8. Antibacterial Activity of Phenolic Acid-g-SSPS Films

The antibacterial activity of phenolic acid-g-SSPS films against *Staphylococcus aureus* and *Escherichia coli* was evaluated [27]. First, the film (40 mg) was soaked in deionized water (4.0 mL, 24 h). The supernatant (1.0 mL) was put in a covered measuring bottle. The 6-mm-diameter filter paper was then immersed in the measuring bottle for 5 min. Then the bacterial suspension (0.2 mL, 10^6^ CFU/mL) was evenly dispersed on an agar plate. The filter paper was removed from the bottle and put on the agar plate to incubate at 37 °C for 1 day. The diameter of the inhibition circle was then recorded with vernier calipers. Sterile water and potassium sorbate were considered the blank and positive control, respectively.

### 2.9. Statistical Analysis

Three parallel tests were performed for each group, and the values were stated as mean ± standard deviation. The values were subjected to Duncan’s multiple range test, and analysis of variance (ANOVA) with SPSS 25.0 software (Armonk, NY, USA), and a significant difference between the values was considered obtained if *p* < 0.05.

## 3. Results

### 3.1. FT-IR Spectra of Phenolic Acid-g-SSPS

As presented in Figure 1, for SSPS, the characteristic peaks at 3326, 2922 and 1610 cm^−1^ were assigned to the O–H stretching vibrations, C–H and COO– stretching vibrations [29]. The peak observed at 1413 cm^−1^ corresponded to the bending vibration of C–H. In addition, some typical peaks of polysaccharides were observed in the range of 1000–1200 cm^−1^, which are due to ring vibrations overlapped with C–OH stretching vibrations and C–O–C glycosidic bond vibrations [30]. The absorption peak at 890 cm^−1^ is assigned to the β-glycosidic linkage in SSPS [31]. The above characteristic absorption bands can also be observed in phenolic acid-g-SSPS, revealing that the phenolic acid-grafted SSPS did not affect the main structure of the SSPS. A new band appeared around 1731 cm^−1^ in phenolic acid-g-SSPS, which may be due to the ester groups (C=O stretching vibrations) formed between the hydroxyl group of SSPS and carboxyl groups of phenolic acid. A new absorption band at 1731 cm^−1^ was also noted in gallic acid-grafted chitosan, indicating the formation of an ester bond [32].

### 3.2. UV-Vis Spectra of Phenolic Acid-g-SSPS

UV-vis spectra of SSPS and phenolic acid-g-SSPS are represented in Figure 2. SSPS did not exhibit an absorption peak within 200–400 nm. PPA-g-SSPS, PA-g-SSPS and GA-g-SSPS showed new absorption peaks around 236, 251 and 265 nm, respectively, which were attributed to the π-system of the benzene ring in phenolic acid [33]. The result indicated that PPA, PA and GA were successfully grafted with SSPS. Researchers have found that PA-g-chitosan showed an absorption peak around 258 nm [12], and GA-g-O-carboxymethyl chitosan had an absorption peak around 269 nm [33].

### 3.3. ^1^ H NMR Spectra of Phenolic Acid-g-SSPS

The ^1^ H NMR spectra can also provide structural support for phenolic acid-g-SSPS in Figure 3. The main signals found in the SSPS spectrum were attributed to the protons of D-galacturonic acid (GalA) at 5.02 (H-1), 3.61 (H-2), 3.90 (H-3), 4.10 (H-4) and 4.36 ppm (H-5) [34], respectively. These signals between 5.06 and 5.19 ppm are owing to the H-1 of various types of arabinose (Ara) [35]. Phenolic acid-g-SSPS showed the proton signals of SSPS. Meanwhile, several new proton signals in the scope of 6.0–8.0 ppm appeared, which should be ascribed to the aromatic protons of the phenolic acid moieties [2,12,36]. The results revealed that phenolic acid-g-SSPS had been prepared.

### 3.4. The Grafting Ratio of Phenolic Acid-g-SSPS

The grafting ratio of phenolic acid-g-SSPS was evaluated by the Folin-Ciocalteu method. The grafting ratio of PPA-g-SSPS, PA-g-SSPS and GA-g-SSPS was determined to be 29.45, 31.76 and 30.74 mg/g, respectively. Studies have shown that the biological activities of phenolic acid-grafted chitosan and the physical, mechanical and biological activities of phenolic acid-grafted chitosan films were related to the grafting ratio and the structure of phenolic acid [2,11]. In our study, three kinds of phenolic acid-g-SSPS with a similar grafting ratio were prepared by controlling the proportion of SSPS and phenolic acid. The aim of our work is to reveal the effect of the structure of phenolic acid on SSPS and SSPS films.

### 3.5. Antioxidant Activity of Phenolic Acid-g-SSPS

#### 3.5.1. DPPH Radical Scavenging Activity

DPPH is extensively used to assess the antioxidant ability of samples because it is a stable free radical [37]. As shown in Figure 4, SSPS showed almost no DPPH radical scavenging ability, but phenolic acid-g-SSPS exhibited stronger DPPH radical scavenging ability compared to SSPS. DPPH’s radical scavenging ability increased in the following order PPA-g-SSPS < PA-g-SSPS < GA-g-SSPS. Phenolic acid-g-SSPS possesses increased DPPH radical scavenging ability, which may be explained by the functional properties of phenolic acid. The antioxidant ability of phenolic acid is inseparable from the position and number of phenolic hydroxyl groups in its molecular composition [38]. PPA, PA and GA possess one, two and three phenolic hydroxyl groups in their molecules, respectively. Therefore, GA-g-SSPS showed the best DPPH radical scavenging ability, which could be owed to its highest number of phenolic hydroxyl groups.

#### 3.5.2. Reducing Power

The reducing power of phenolic acid-g-SSPS is presented in Figure 5; the high absorbance indicates the good reducing power. SSPS showed almost no absorbance, which revealed that SSPS has no reducing power. After phenolic acid was grafted with SSPS, the reducing power of SSPS was improved to varying degrees, indicating that the reducing power of the copolymer mainly came from phenolic acid [39]. The absorbance of GA-g-SSPS and PA-g-SSPS was 0.375 and 0.203 at 2.5 mg/mL, indicating that the reducing power of GA-g-SSPS and PA-g-SSPS was enhanced, but the reducing power of PPA-g-SSPS was not significantly improved. GA-g-SSPS exhibited the highest reducing power. The result implied that the number of phenolic hydroxyl groups of phenolic acid is positively correlated with the reducing power of phenolic acid-g-SSPS.

### 3.6. Properties of Phenolic Acid-g-SSPS Films

#### 3.6.1. Thickness

As seen in Table 1, the thickness of all films changed from 60 to 69 µm. The result indicated that the phenolic acid-grafted SSPS had little effect on the thickness of the SSPS film; other researchers have reported similar results [14,15].

#### 3.6.2. Moisture Content and Water Solubility

The water resistance of a film can be assessed by two indicators, water solubility and moisture content. The moisture content of phenolic acid-g-SSPS films was higher than that of SSPS film (Table 1). It might be due to the fact that the benzene ring of the phenolic acid hindered the free volume of the film, which further resisted the diffusion of water molecules [40]. The water solubility is another important factor of the film since high water solubility is not conducive to the preservation of stored food. The water solubility of phenolic acid-g-SSPS films was much lower in comparison to that of SSPS films (Table 1). It might be explained by the fact that the interactions among the groups of gelatin, phenolic acid, glycerol and SSPS reduced the usability of hydroxyl groups, interfering with the interaction between SSPS and water [25]. Moreover, the water solubility of phenolic acid-g-SSPS films was reduced in the following order, PPA-g-SSPS film > PA-g-SSPS film > GA-g-SSPS film. The finding may be caused by the covalent cross-linking of phenolic acid with SSPS, resulting in the formation of long-chain molecules with low water solubility [25].

#### 3.6.3. Water Vapor Permeability

A key role of food packaging film is to prevent or reduce as much as possible, the transfer of moisture from the environment to the food [41]. Thus, low WVP facilitates an optimized food packaging environment and has the possibility of extending the shelf life of food. The WVP of SSPS and phenolic acid-g-SSPS films are listed in Table 1. Phenolic acid-g-SSPS films exhibited relatively lower WVP than SSPS film, varying from 3.44 × 10^−7^ to 3.70 × 10^−7^ g mm/h cm^2^ Pa. The WVP of PA/GA-grafted CS films was lower than that of CS films [14,15]. Covalent bonds between phenolic acid and SSPS greatly restricted the availability of hydrophilic areas of SSPS, thereby reducing the hydrophilicity of phenolic acid-g-SSPS films [42]. In addition, the reduced WVP in phenolic acid-g-SSPS films can be ascribed to the heavy benzene ring hindering the intra- and inter-molecular hydrogen bonding system of SSPS films [15].

#### 3.6.4. Mechanical Property

Enough mechanical strength can guarantee the integrity of the film and protect the film against small flaws, so TS and EB are two key indicators of the film’s mechanical properties [43]. TS and EB of phenolic acid-g-SSPS films were lower than those of SSPS film (Table 1). This may be due to the covalent bond between phenolic acid and SSPS, which makes it more difficult to form intermolecular hydrogen bonds between SSPS and other components of the film (gelatin, glycerol) [36,44]. The GA-g-SSPS film exhibited the worst mechanical properties, which could be explicated by the three phenolic hydroxyl groups in the GA-g-SSPS molecule. Excessive phenolic hydroxyl groups could disrupt the inner structure of the film and reduce the intermolecular forces between polymer chains, resulting in a decrease in the TS and EB of the film [15].

### 3.7. The released Phenol Content and DPPH Radical Scavenging Ability

Table 2 shows the DPPH radical scavenging ability and the released phenol content of the films. The released phenol content from the SSPS, PPA-g-SSPS, PA-g-SSPS and GA-g-SSPS film are 1.95, 6.16, 8.45 and 12.05 mg/g, respectively. The result revealed that after phenolic acid was grafted with SSPS, the released phenol content increased, and the released phenol content from the phenolic acid-g-SSPS films increased with the increase in the number of phenolic hydroxyl groups in the phenolic acid. Phenolic acid-g-SSPS films show DPPH radical scavenging ability, but SSPS films almost do not possess the ability, suggesting that phenolic acid endowed the SSPS film with increased antioxidant ability. Moreover, the order of DPPH radical scavenging ability is as follows: GA-g-SSPS film > PA-g-SSPS film > PPA-g-SSPS film. The result indicates that the DPPH radical scavenging ability is positively linked with the released phenol content. The DPPH radical scavenging ability of GA-grafted CS film is better than that of PA-grafted CS film [2].

### 3.8. Antibacterial Activity

As listed in Table 2, phenolic acid-g-SSPS films exhibited stronger antibacterial activity against *Staphylococcus aureus* and *Escherichia coli* than SSPS film. The order of antibacterial ability is as follows: GA-g-SSPS film > PA-g-SSPS film > PPA-g-SSPS film. The result exhibits that the antibacterial ability of phenolic acid-g-SSPS films is positively correlated with the released phenol content. The GA-g-SSPS film that has the highest released phenol content possesses the best antibacterial ability. In addition, due to differences in the structure of the bacterial outer membrane, all films exhibit stronger inhibitory ability against Staphylococcus aureus than that of Escherichia coli [45]. Compared with Gram-negative bacteria, GA-grafted chitosan film showed stronger antibacterial ability against Gram-positive bacteria [15].

## 4. Conclusions

In this work, PPA, PA and GA were grafted onto SSPS by the EDC/NHS coupling method through ester bonding, respectively. FT-IR, UV-vis and ^1^ H NMR spectroscopy revealed that the reaction had successfully occurred. Phenolic acid endowed the SSPS with antioxidant properties. GA-g-SSPS possessed the best DPPH radical scavenging ability and reducing power, which could be linked to the three phenolic hydroxyl groups in GA. Meanwhile, phenolic acid-g-SSPS can decrease the water solubility and WVP and enhance the biological activities of the SSPS film. Since GA-g-SSPS film showed the best antioxidant and antibacterial performances, it can be expected to be a progressive material for food-active packaging. In subsequent studies, the influence of GA-g-SSPS film on the freshness of foods such as vegetables, fruits and meats should be assessed. The results in this study have important insights into the performance of active food packaging materials and their possible applications for the future.

## Figures and Tables

**Figure 1 foods-11-03747-f001:**
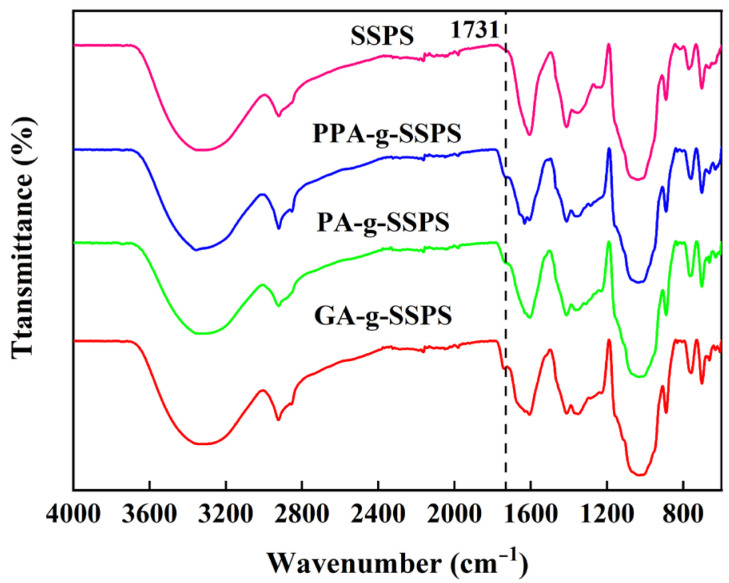
FT-IR spectra of phenolic acid-g-SSPS.

**Figure 2 foods-11-03747-f002:**
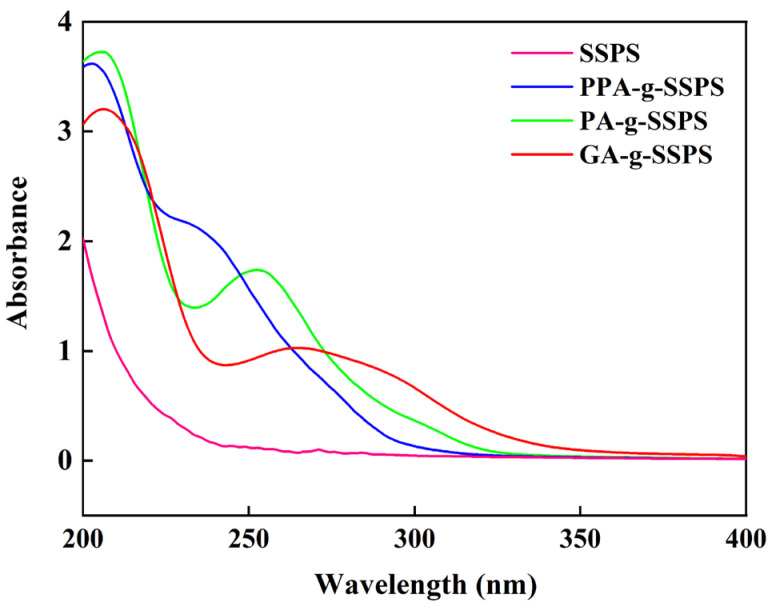
UV-vis spectra of phenolic acid-g-SSPS.

**Figure 3 foods-11-03747-f003:**
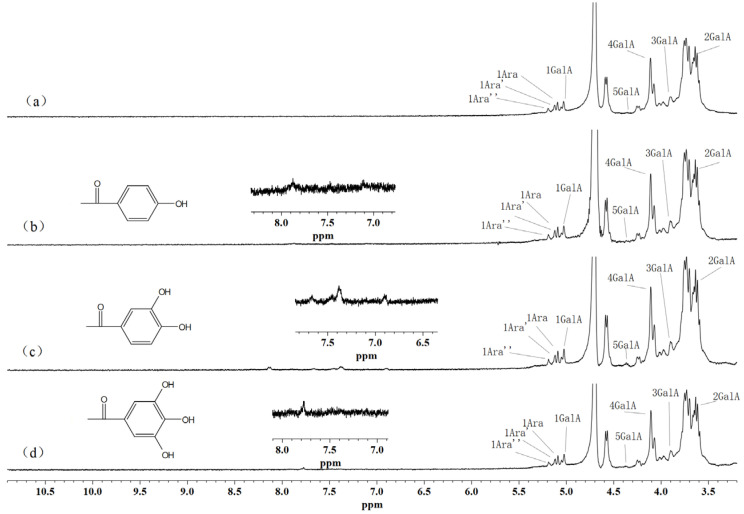
^1^ H NMR spectra of SSPS (**a**), PPA-g-SSPS (**b**), PA-g-SSPS (**c**) and GA-g-SSPS (**d**).

**Figure 4 foods-11-03747-f004:**
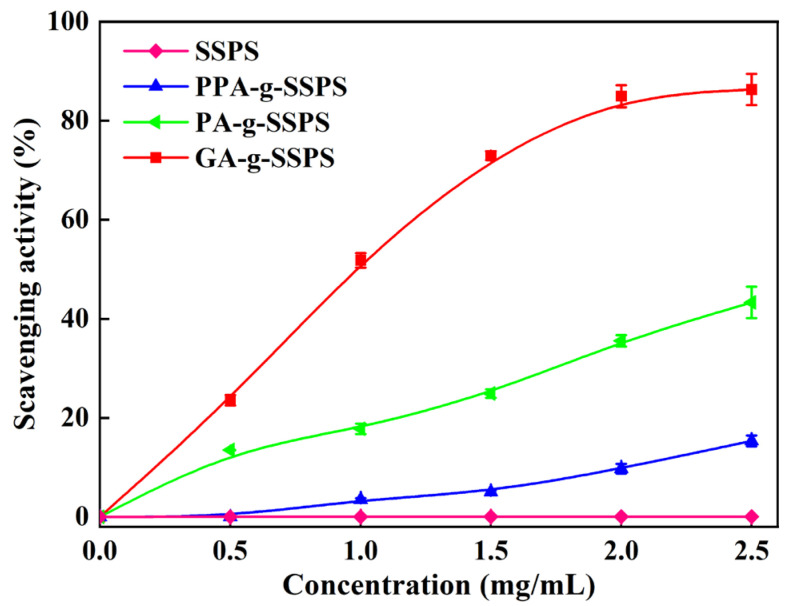
DPPH radical scavenging ability of phenolic acid-g-SSPS.

**Figure 5 foods-11-03747-f005:**
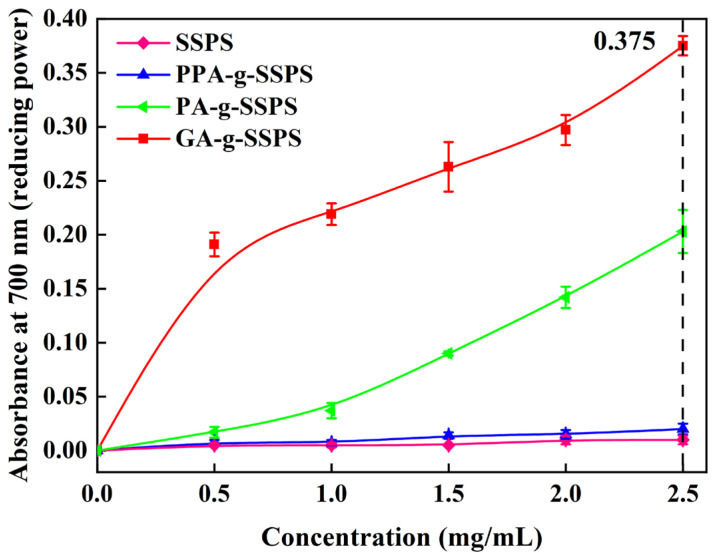
Reducing power of phenolic acid-g-SSPS.

**Table 1 foods-11-03747-t001:** Physical and mechanical properties of phenolic acid-g-SSPS films.

Films	Film Thickness (µm)	Moisture Content (%)	Water Solubility (%)	Water Vapor Permeability (×10^−7^ g mm/h cm^2^ Pa)	Tensile Strength (MPa)	Elongation at Break (%)
SSPS film	60 ± 11	10.62 ± 0.21 ^a^	98.57 ± 0.35 ^d^	4.04 ± 0.059 ^d^	30.08 ± 1.54 ^d^	15.1 ± 1.68 ^d^
PPA-g-SSPS film	62 ± 3	11.36 ± 0.14 ^b^	90.59 ± 1.36 ^c^	3.44 ± 0.032 ^a^	28.05 ± 1.33 ^c^	13.2 ± 1.12 ^c^
PA-g-SSPS film	65 ± 6	11.00 ± 0.25 ^a^	87.58 ± 0.38 ^b^	3.66 ± 0.047 ^b^	24.57 ± 2.14 ^b^	12.5 ± 2.16 ^b^
GA-g-SSPS film	69 ± 5	10.39 ± 0.21 ^a^	81.60 ± 1.27 ^a^	3.70 ± 0.051 ^c^	23.35 ± 1.77 ^a^	12 ± 1.68 ^a^

Values are given as mean ± standard deviation (SD). Different letters in the same column indicate significantly different (*p* < 0.05).

**Table 2 foods-11-03747-t002:** Released phenol content, DPPH radical scavenging and antibacterial activities of phenolic acid-g-SSPS films.

Films	Released Phenol Content (mg/g)	DPPH Radical Scavenging Activity (%)	Inhibition Zone (mm)	
			*Staphylococcus aureus*	*Escherichia coli*
SSPS film	1.95 ± 0.47 ^a^	0.31 ± 0.11 ^a^	6.35 ± 0.12 ^b^	6.24 ± 0.18 ^b^
PPA-g-SSPS film	6.16 ± 0.72 ^b^	5.94 ± 0.17 ^b^	7.91 ± 0.27 ^c^	7.28 ± 0.32 ^c^
PA-g-SSPS film	8.45 ± 1.33 ^c^	63.62 ± 1.21 ^c^	8.77 ± 0.29 ^d^	8.36 ± 0.16 ^d^
GA-g-SSPS film	12.05 ± 1.77 ^d^	81.47 ± 1.78 ^d^	10.20 ± 0.36 ^e^	9.59 ± 0.72 ^e^
Potassium sorbate	-	-	10.12 ± 0.21 ^f^	9.87 ± 0.24 ^e^
Sterile water	-	-	6.00 ± 0.00 ^a^	6.00 ± 0.00 ^a^

Values are given as mean ± standard deviation (SD). Different letters in the same column indicate significantly different (*p* < 0.05).

## Data Availability

Data is contained within the article.

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
