# Peer review of "Physical, Mechanical and Biological Properties of Phenolic Acid-Grafted Soluble Soybean Polysaccharide Films"

_foods, 2022, doi:10.3390/foods11223747_

Round 1
Reviewer 1 Report
foods-2009443-peer-review-v1-1
Physical, mechanical and biological properties of phenolic acid 2 grafted soluble soybean polysaccharide films
by Mengyang Zhang et al.
The authors report on the influence of different phenolic acids on the properties of films of such grafted soluble soybean polysaccharides. However, there are numerous issues that must be reconsidered by the authors.
The entire manuscript must be checked for singular/plural terms together with the proper use of was/were.
L41: “phenolic acids are frequently grafted with chitosan” should be “chitosan was frequently grafted with…” By the way, is this a grafting reaction? Is there any hint for side-chain growth?
L44: “dramatic” should be replaced by “drastic”
L63: Folin-Ciocalteu-reagent
L66ff: stoichiometric/molar amounts of the reagents and the yield must be given. Otherwise, the efficacy of the conversions cannot be assessed.
L71: mixed with aqueous ethanol (20 mL, 70 %, v/v)
L75: the solution was stirred at room temperature
L84f: FT-IR spectra of … were recorded on a … FT-IR spectrometer
L116: was allowed to stand for 0.5 h
2.6.4: What about statistics?
L182ff: There are “bands” in the FT-IR spectra but no peaks
Unfortunately, the conversion itself is not further discussed here except mentioning the grafting ratio at P6.
L197: which are attributed to
Figure 3 must contain the substituent but not the carboxylic acid; The peaks in the aromatic region appear very weak. What is the accuracy of the integration, hence, the calculation of the grafting ratio?
Figure 4: The color-code for samples SSPS and GA-g-SSPS should be chosen to be more different.
L247-L249: Is this expected or unexpected?
The films are water soluble. Is this property favorable for packaging applications?
L291ff: Maybe I did not get the point correctly: What is the reason for phenol release? Aren’t the carboxylic acid esters unstable? It is not really clear whether or not the carboxylic acids are covalently bound or physically mixed with the biopolymer.
L306: species names should be typed in italics
Reviewer 2 Report
Dear Dr.
This manuscript described the Physical, mechanical and biological properties of phenolic acid grafted soluble soybean polysaccharide films. At first glance, the topic is interesting and the novelty is good. However, the minor remarks are listed below:
The abstract needs to be improved to be a sharp and standalone. The current did not represent the observed values, please keep reminded that abstract should represent all sections of manuscript.
The unit of Film thickness results changed to µm.
Why control film (SSPS film) had released phenol content?
Best regards
Author Response
Dear Reviewer,
Thank you very much for your critical comments and thoughtful suggestions on our manuscript (Manuscript ID: foods-2009443) entitled “Physical, mechanical and biological properties of phenolic acid grafted soluble soybean polysaccharide films”. We had studied your comments carefully and had made corrections which we hope to meet with your approval. The changes had been highlighted by using red-colored text in the manuscript.
Below we provide our point-by-point explanations to the reviewer’s comments/questions:
Point 1: The abstract needs to be improved to be a sharp and standalone. The current did not represent the observed values, please keep reminded that abstract should represent all sections of manuscript.
Response 1: Thank you very much for your useful suggestion. The abstract had been modified and the related information had been supplemented in line 12-14, line17-19 and line 21-26.
Point 2: The unit of Film thickness results changed to µm.
Response 2: Thank you very much for your suggestion. The expression had been changed in line 273 and in Table 1.
Point 3: Why control film (SSPS film) had released phenol content?
Response 3: Thank you for your meaningful question. Liu’s work showed that the released phenol content of chitosan film was 3.31 mg/g, which was probably due to the presence of chromogens [1]. The released phenol content of soluble soybean polysaccharide film was about 3 mg/g in Salarbashi’s study [2]. But the reason is not clear and needs further investigation.
References
[1] Liu, J.; Liu, S.; Wu, Q.Q.; Gu, Y.Y.; Kan, J.; Jin, C.H. Effect of protocatechuic acid incorporation on the physical, mechanical, structural and antioxidant properties of chitosan film. Food Hydrocoll. 2017, 73, 90-100. https://doi.org/10.1016/j.foodhyd.2017.06.035
[2] Salarbashi, D.; Tajik, S.; Shojaee-Aliabadi, S.; Ghasemlou, M.; Moayyed, H.; Khaksar R.; Noghabi, M.S. Development of new active packaging film made from a soluble soybean polysaccharide incorporated Zataria multiflora Boiss and Mentha pulegium essential oils. Food Chem. 2014, 146, 614-622. https://doi.org/10.1016/j.foodchem.2013.09.014
